# Earlier preterm birth is associated with a worse neurocognitive outcome in a rabbit model

**Johannes van der Merwe**[1,2], **Lennart van der Veeken**[1,2], **Analisa Inversetti**[1], **Angela Galgano**[1], **Jaan Toelen**[1,3], **Jan Deprest**[1,2,4]*

**1** Department of Development and Regeneration, Cluster Woman and Child, Group Biomedical Sciences, KU Leuven University of Leuven, Leuven, Belgium, **2** Division Woman and Child, Department of Obstetrics and Gynaecology, University Hospitals Leuven, Leuven, Belgium, **3** Division Woman and Child, Department of Pediatrics, University Hospitals Leuven, Leuven, Belgium, **4** Institute for Women's Health, University College London, London, United Kingdom

* jan.deprest@uzleuven.be

**Data Availability Statement:** All relevant data are within the manuscript and Supporting Information files.

## Abstract

### Background

Preterm birth (PTB) and particularly late preterm PTB has become a research focus for obstetricians, perinatologists, neonatologists, pediatricians and policy makers alike. Translational models are useful tools to expedite and guide clinical but presently no model exists that contextualizes the late PTB scenario. Herein we aimed to develop a rabbit model that echo's the clinical neurocognitive phenotypes of early and late PTB.

### Methods

Time mated rabbit does underwent caesarean delivery at a postconceptional age (PCA) of either 28 (n = 6), 29 (n = 5), 30 (n = 4) or 31 (n = 4) days, term = 31 d. Newborn rabbits were mixed and randomly allocated to be raised by cross fostering and underwent short term neurobehavioral testing on corrected post-natal day 1. Open field (OFT), spontaneous alteration (TMT) and novel object recognition (NORT) tests were subsequently performed at 4 and 8 weeks of age.

### Results

PTB was associated with a significant gradient of short-term mortality and morbidity inversely related to the PCA. On postnatal day 1 PTB was associated with a significant sensory deficit in all groups but a clear motor insult was only noted in the PCA 29d and PCA 28d groups. Furthermore, PCA 29d and PCA 28d rabbits had a persistent neurobehavioral deficit with less exploration and hyperanxious state in the OFT, less alternation in TMT and lower discriminatory index in the NORT. While PCA 30d rabbits had some anxiety behavior and lower spontaneous alteration at 4 weeks, however at 8 weeks only mild anxiety driven behavior was observed in some of these rabbits.

**Funding:** JvdM and LvdV are funded with support of the Erasmus + Programme of the European Union (Framework Agreement number: 2013-0040). This publication reflects the views only of the author, and the Commission cannot be held responsible for any use which may be made of the information contained therein. The funders had no role in study design, data collection and analysis, decision to publish, or preparation of the manuscript.

**Competing interests:** The authors have declared that no competing interests exist.

## Conclusions

In this rabbit model, delivery at PCA 29d and PCA 28d mimics the clinical phenotype of early PTB while delivery at PCA 30d resembles that of late PTB. This could serve as a model to investigate perinatal insults during the early and late preterm period.

## Introduction

Preterm birth (PTB), defined as a birth before 37w0d of gestation, is worldwide the second most common cause-of-death in children younger than 5 years [1]. Additionally, for neonates who survive, PTB is the leading contributor to many substantial long-term impairments [2]. Obviously the earlier the baby is born, the greater these risks will be especially for cognitive, motor and behavioral performances [3]. In view of the adverse sequalae of PTB, it has been suggested that mothers at risk of PTB and their ensuing preterm infants could benefit from more careful consideration and direct management. A broad consensus to avoid iatrogenic PTB is easily accepted by health care providers and readily enforced by most organizations [4]. Beyond this starting point, current clinical management is mainly directed at optimizing the outcomes of early preterm birth, i.e. birth before 34w0d [5].

Yet the late-preterm infant, defined as a birth at 34w0d to 36w6d of gestation, is also faced with a wide array of neonatal risks [6–8] and significant behavioral and emotional difficulties are already evident at school age [9]. The burden of late PTB becomes even more obvious in view of the fact that late PTB accounts for about 70% of all PTB [5, 10]. This underpins the reason why late PTB has become a research focus for obstetricians, perinatologists, neonatologists, pediatricians and policy makers alike. Current risk reducing clinical strategies, especially by antenatal management, is mainly guided by suboptimal quality research, evidence from early PTB and expert opinion [5]. Many research questions typically encountered in this late preterm period persists with vague answers. Appropriate translational research models could serve as a useful tool to expedite and guide clinical research.

Translational research plays a critical role in testing hypotheses encountered through clinical observations in a setting where certain variables could be controlled. As with most pathophysiological processes not one specific animal model could be sufficient to replicate the complex physiology of human gravidity and fetal development. In terms of PTB, variations in the animal's inherent maternal physiology, fetal maturation and parturition all play a role in determining the model's validity [11]. A recent review noted the inconsistencies within PTB models and especially the overuse of hypoxic-ischemic or infective/inflammatory mediated insults [12]. Presently no model exists that specifically contextualizes the late PTB scenario.

The rabbit has been widely employed in perinatal neurobehavioral research, simulating cerebral palsy [13], hypoxia-ischemia [14], intraventricular hemorrhage [15] and intrauterine infection [16]. The rabbit seems best suited to bridge the gap between small and large animals being widely available, having low housing needs, a short reproductive cycle, timed gestation with large litters and placental development close to human [17]. In translational neuroscience rabbits have the advantage of a more complex brain structure with greater white matter proportion than rodents and the timing of perinatal brain white matter maturation is comparable to the human, hence they suffer from the same vulnerability as humans [18]. We have already characterized the early neuropathological consequences of iatrogenic preterm birth in the rabbit model [19]. Therein delivering the rabbit at a postconceptional age (PCA) of 28 days (term 31 days) resembles early severe PTB with a clear neuropathological phenotype on corrected postnatal day 1.

Since the aforementioned models mainly used insults beside prematurity and reported mostly on short-term neuropathological outcomes, we wanted to characterize the short- and long-term neurobehavioral spectrum of iatrogenic prematurity in a rabbit model. Ultimately to define at which PCA iatrogenic prematurity resembles the clinical phenotype of early and late PTB.

## Materials and methods

### Animal delivery and care

Animals were treated according to current guidelines for animal well-being and research [20], and all experiments were approved by the Ethics Committee for Animal Experimentation of the Faculty of Medicine (P062/2016 and P112/2017). Time-mated pregnant rabbit does (*Oryctolagus cuniculus*; Hybrid of New Zealand White/Black and Flemish Giant Rabbit) were housed in separate cages before delivery at 21°C, 42% humidity, with a 12-hour day-night cycle and free access to water and food. The does underwent a caesarean delivery at a PCA of either 28 (n = 6), 29 (n = 5), 30 (n = 4) or 31 (n = 4) days, hereafter indicated by PCA 28d, PCA 29d, PCA 30d and PCA 31d respectively. The New Zealand white rabbit has a gestation period of 31 days and delivery before PCA 28 days invariably results in death [21]. Rabbits were allocated to short term, corrected postnatal day1, or long-term, 4 and 8 weeks of age, evaluations. All animals were inspected daily, weighed daily for the first week and then weighed weekly thereafter. Experimental setup depicted in Fig 1.

Before delivery, does were weighed and premedicated with intramuscular ketamine (15 mg/kg, Nimatek®; Eurovet Animal Health BV, Bladel, The Netherlands) and medetomidine (25 mg/kg, Domintor®, Orion Pharma, Aartselaar, Belgium). The doe was placed in supine position and local anesthesia (2% lidocaine hydrochloride, Xylocaine®, AstraZeneca, Brussel, Belgium) was injected before the skin incision. Following delivery, the doe was euthanized with a mixture of 200 mg embutramide, 50 mg mebezonium, and 5 mg tetracain hydrochloride (intravenous bolus of 1 mL T61®; Intervet International BV, Boxmeer, The Netherlands).

At delivery, the kittens were dried, stimulated, weighed and placed in an incubator (TLC-50 Advance, Brinsea® Products, Weston Super Mare, UK) at 32°C with 60% humidity as described [19]. Afterwards a subcutaneous microchip was implanted in the scruff for identification (Compact Max Datamars, AgnTho's AB, Lidingö, Sweden) and fed once, via a 2.5 Fr orogastric tube, with a milk replacer containing 30% proteins and 50% of lipids (FoxValley 30/50, Lakemoor Illinois, US), Bio-Lapis for electrolytes, vitamins and probiotics (Protexin

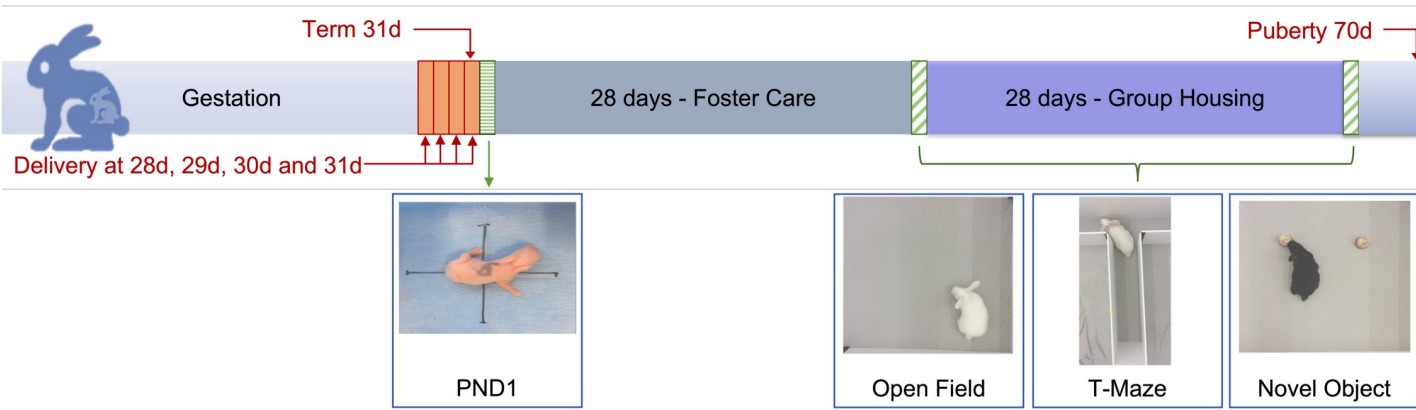

**Fig 1. Study setup.** Does were electively delivered at a postconceptional age of 28, 29, 30 or 31 days. Neurobehavioral assessments were done on corrected postnatal day 1, at 4 weeks and 8 weeks of age.

Veterinary, Somerset, UK) and Col-o-cat for a high amount of immunoglobulins (Sanobest, 's Hertogenbosch, Netherlands). Kittens were then randomized with Research Randomizer (http://www.randomizer.org) to either a short term, evaluation at PCA 32 days, or long term, evaluation at 4 and 8 weeks, group.

Kittens were mixed and randomly allocated to be raised by cross fostering in groups of 8–10, by a doe that did not undergo any kind of manipulation and spontaneously delivered the day before. The foster mother was housed in a double cage to which a nesting box (54 × 31 × 27 cm) was connected. The foster mother raised the kittens for a period of 4 weeks until they were fully weaned, whereafter they were housed in groups of 4–6 rabbits in a double cage at 21˚C, 42% humidity, with a 12-hour day-night cycle and free access to water and food until 8 weeks post-delivery.

## Neurobehavioral testing

Neurobehavioral evaluations were carried out in the mornings and done in a designated space with auditory and olfactory contamination kept to a minimum. The animals' health was assessed before neurobehavioral testing was done, by checking the animal's general health and weight. All videos were recorded, anonymized and scored by an observer blinded to the group allocation.

In the short-term group the PND1 was performed on corrected postnatal day 1, i.e. PCA of 32 days.

**Postnatal day 1 evaluation, PND1.** PND1 was performed in a designated 30 × 30cm area. Before handling they were left undisturbed in this assessment area and thereafter video-taped for 90 seconds. During this period the motor function (tone, motor activity, and loco-motion and gait) were assessed and afterwards the sensation (touching the whiskers with cotton swab and a mild pin prick to evaluate pain sensation on the hind limbs) including cranial nerves (olfaction, sucking and swallowing and head turn to feeding and righting reflex) were assessed, as previously described [19]. (S1 File)

The sum of these scores represented the total neuromotor (maximum score 26) and neuro-sensory evolution (maximum score 15).

In the long-term group the open field, T-maze and novel object recognition tests were performed on a postnatal age of 4 and 8 weeks. This was done in a designated room that was insulated from sound, temperature controlled at 21˚C and with full overhead illumination at 300 lux. The area was cleared of any urine and feces after each test. Rabbits were handled daily and habituated in the test room the day before the testing was started. For habituation, the animals were placed in the open field for 30 min without any objects or interference. The sequence of assessments was habituation on day 1, open field on day 2, T-maze on day 3 and 4 and novel object recognition test on day 5. (S1 File)

**Open Field Test, OFT.** The OFT served as an evaluation of emotional reactivity and anxiety, exploratory- and non-exploratory locomotion. The OFT was designed and used in accordance with the procedure previously described [22] and was performed on a designated 80 × 80 cm with 80cm-high polyvinyl chloride walls and a rubber floor divided into nine numbered squares. Each rabbit was individually placed in a starting box (20 × 20 cm at 4 weeks; 25 × 25 cm at 8 weeks), which was removed after 60 seconds. The animal was then left free to explore the open field over a period of 5 minutes. Afterward it was returned to its cage. The following behaviors were scored:

- Latency time of leaving the starting point (seconds)

- Total and central displacements, number of squares crossed (n)

- Running episodes (n)

- Escape attempts (n)

- Exploration events, moving with forelegs or standing while sniffing and looking around inside the same square (n)

- Hops, the number of times the rabbit completely displaced its body by a hop (n)

- Standing still, with fore and hind legs not stretched and, on the ground (n)

- Resting, totally inactive with the body touching the floor and fore and/or hind legs stretched on the ground (n)

- Rearing, number of times the rabbit upheaves on its hind legs (n)

- Self-grooming events (n)

- Digging events (n)

- Biting, biting any elements of the pen (n)

- Defecation and urination (n)

**Novel Object Recognition Test, NORT.** The NORT served as an evaluation of the rabbit's response to novelty, object-directed exploration in order to indirectly to assess recognition memory and declarative memory. The NORT was performed in the OFB pen and was adapted from the original description [23] including some modifications in the stimulus used. Instead of using visual stimulus, an odor-based stimulus was used by means of placing pieces of fruit (apple or orange) inside perforated plastic containers, since olfactory sensitivity is highly developed in rabbits [24]. This is in agreement with the notion that the type of stimulus presented must be one in which the sensory perception of the species chosen is adequate.

Each session was composed of 3 phases. During the 'sample phase', the rabbit was allowed to explore the pen for 5 minutes with two containers that had the same odor-based stimulus (apple). Afterward, in the 'retention phase' the rabbit was returned to the transport box for a 30-minute interval. Finally, during the 'testing phase', 2 different odor-based stimuli (apple and orange) were presented to the animal for 5 minutes more. At the end of the test, the rabbit was returned to its cage.

For both phases, exploration of the object was considered when the rabbit showed sniffing, touching, and having moving vibrissae while directing the nose toward the object at a distance of less than 1 cm.

Cumulative events exploring each object in the two sessions was recorded. Finally, the discrimination index (DI), which represents the ability to discriminate the novel from the familiar object, will be calculated as follows:

$$DI = \frac{(Novel\ Object\ Exploration\ Time - Familiar\ Object\ Exploration\ Time)}{(Novel\ Object\ Exploration\ Time + Familiar\ Object\ Exploration\ Time)}$$

**T-Maze Test, TMT.** The TMT was used to assess the rabbits spatial learning and short-term memory and free-choice exploration. The TMT was performed in accordance with published methods for rodents [25] and modified for rabbits [26]. Each rabbit was tested once daily for 2 consecutive days.

The maze was composed of a common corridor and 2 goal arms (80 × 15 cm at 4 weeks and 80 × 20 cm at 8 weeks). The 2 goal arms were not separated, so the animal could freely choose either of them and at the end of each arm a black triangle or square was placed as a visual cue.

Each testing session was composed of a sample run and a testing run, with a 15-minute interval in between.

The rabbit was placed in the starting box for 30 seconds before it was allowed to choose a goal arm. Once the animal had entered the common corridor, the starting box was closed. We considered that it had made a choice when all 4 paws were placed in the arm. At this point, the chosen arm was closed to prevent the access to the opposite one, and the rabbit was allowed to explore it for 30 seconds. If the animal failed in the sample run, that is, it did not choose any arm within 3 minutes from the starting box opening, a second sample run was allowed after 15 minutes. The testing run occurred with the same conditions described for the sample run, but no second testing run was allowed in case of failure.

For this test, we evaluated the time spent in the starting area, total spontaneous alternation (expressed as %), and the failure rate in the 2 testing sessions.

## Euthanasia and tissue collection

After neurobehavioral testing on PND1 or after 8 weeks the rabbits were anesthetized with intramuscular ketamine (35 mg/kg, Nimatek®; Eurovet Animal Health BV, Bladel, The Netherlands) and xylazin (6 mg/kg, XYL-M®; VMD, Arendonk, Belgium) and transcardially perfused with 0.9% saline and heparin (100u/mL) followed by perfusion fixation with 4% paraformaldehyde (PFA) in 0.1 mol/L phosphate buffer (pH 7.4). Whole brain volumes, including cerebellum, were determined using a fluid displacement method [27].

## Statistics

Preterm rabbits were expected to perform worse on neurobehavioral testing. Since no previous data existed a priori power calculation could not be calculated but a post hoc power calculation utilizing a one-way ANOVA analysis was used.

Data was analyzed using Prism for Windows version 8.0 (Graphpad software, San Diego, CA, USA). Data was checked for normality of distribution using a D'Agostino-Pearson omnibus normality test, then presented as a mean with standard deviations or median and interquartile ranges. Comparison was done by unpaired students t-test or Mann Whitney test. Group means were compared by one-way ANOVA or Kruskal Wallis test. A p-value <0.05 was considered significant. Adjustment for multiple testing was applied using the Benjamini–Hochberg false discovery rate set at 1% when more than 5 comparisons where done. A Grubbs' test with an α of 0.05 was used to identify outliers. Survival curves are presented as a Kaplan-Meier graph with groups comparisons using a Mantel-Cox test.

## Results

In the PCA31d, PCA30d and PCA28d groups there were no differences in the maternal characteristics or pregnancy outcomes (S1 Table).

### Early mortality and morbidity correlates linearly with the severity of prematurity

PTB was associated with a significant short-term mortality, i.e. with a survival on postnatal day 1 of 100%, 94%, 71% and 56% for PCA31d, PCA30d, PCA29d and PCA28d respectively. In the PCA28d and PCA29d groups there was an ongoing mortality noted up to 21 days of age that ultimately resulted in an 8 week survival of 59% and 37% for these two groups. Both the PCA31d and PCA30d groups had a survival of 90% or more at 8 weeks of age (Fig 2A).

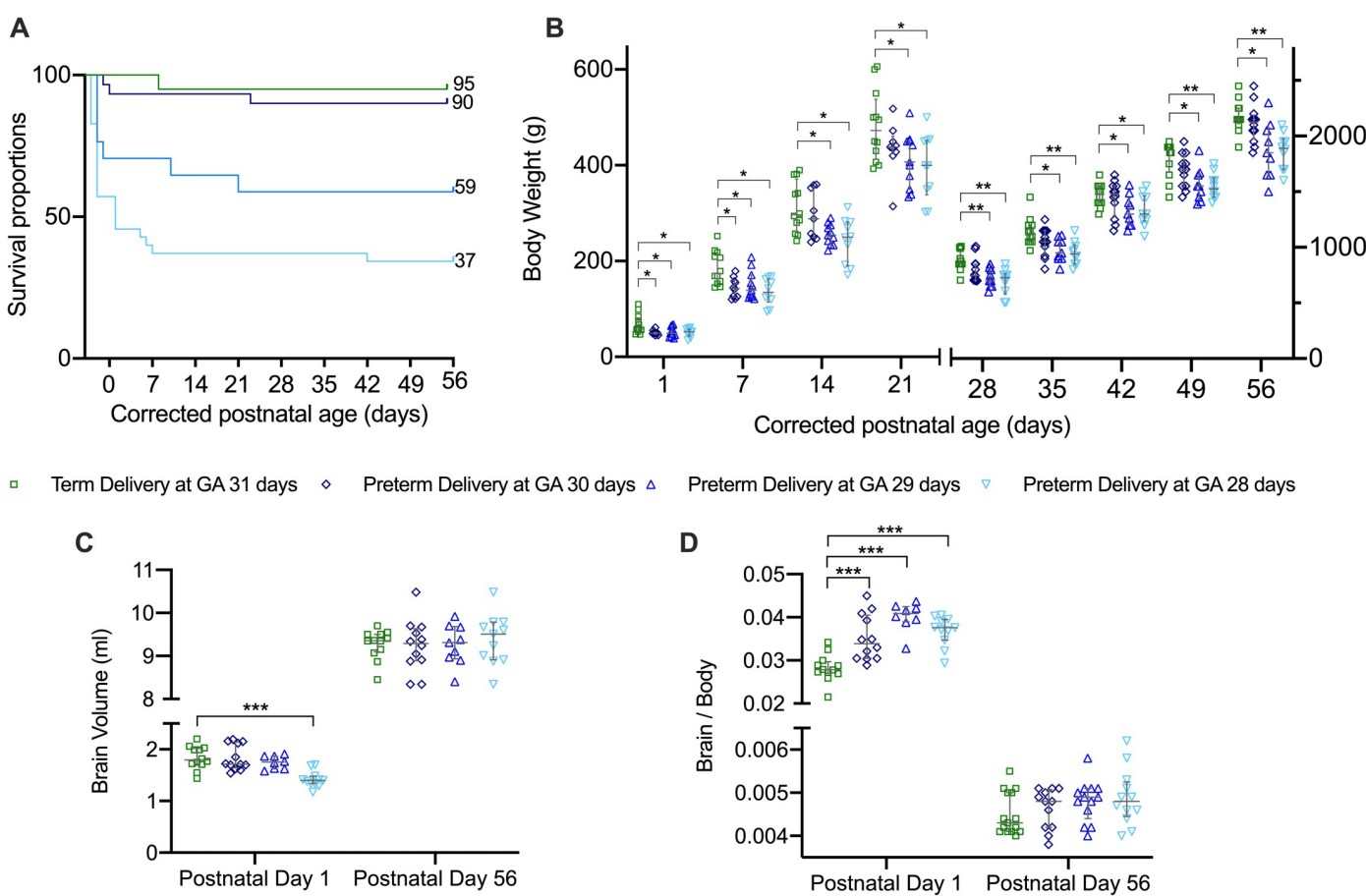

**Fig 2. Rabbits survival and biometrics.** New-born rabbit survival (A), body weight (B), brain weight (C) and brain to body ratios (D). PCA31d n = 12, PCA30d n = 12, PCA29d n = 9 and PCA28d n = 11. Data displayed as median and IQR with significance compared to the term birth group indicated as * 0.05 ≥ p > 0.01; ** 0.01 ≥ p > 0.001; ***p < 0.001.

The PCA31d had a significant higher weight difference from all the other 3 groups up to 14 days of age. Thereafter no difference was seen with the PCA30d group but the PCA29d and PCA28d groups had a persistently lower body weight up to 8 weeks when compared to the PCA31d group (Fig 2B). Likewise there was a tendency to lower brain volumes in the preterm groups, but this was only significantly lower in the PCA28d group at postnatal day 1. In all the preterm groups there was an initial higher brain to body ratio at postnatal day 1, but by 8 weeks of age no differences could be noted anymore (Fig 2C and 2D).

## PTB is associated with a significant short-term neurobehavioral impact

Overall PTB led to an overall sensory deficit noted in all groups on postnatal day 1, that was predominantly due to a pain and touch sensation deficit. There was also a significant motor deficit noted in the PCA29d and PCA28d groups, wherein a clear gait and locomotion deficit was noted (Fig 3 and S2 Table). The severity of the early neurocognitive insult was directly correlated to the PCA of delivery.

## Earlier PTB is associated with a severe and persistent neurobehavioral deficit

In both the PCA29d and PCA28d groups a persistent neurocognitive impairment was demonstrated up to the final assessment at 8 weeks of age. For these two groups there was less central

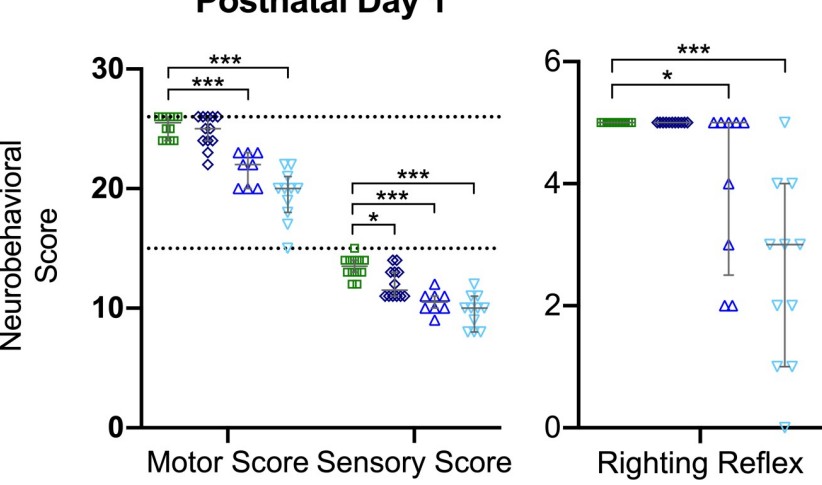

**Fig 3. Neonatal rabbits' neurobehavioral assessment on corrected postnatal day 1.** Total neuromotor and neurosensory score (A) and righting reflex events (B). PCA31d n = 12, PCA30d n = 12, PCA29d n = 8 and PCA28d n = 11. Data displayed as median and IQR with significance compared to the term birth group indicated as * $0.05 \geq p > 0.01$; ** $0.01 \geq p > 0.001$; *** $p < 0.001$.

displacement, less exploration, more rearing events and more biting events noted in the OFT assessments (Table 1). This behavior was indicative of a hyperanxious state with less exploratory tendencies. Furthermore, both at 4 and 8 weeks of age an impact was noted in the short term and spatial memory of these two groups as seen in the lower spontaneous alteration scores in the TMT (Fig 4). Furthermore, in the NORT a persistent lower exploration behavior with lower discriminatory indexes was noted at both 4 and 8 weeks of age in these two early preterm groups (Fig 4). Moreover, at 4 weeks of age there were also less interactions noted in the NORT (S3 Table).

## Later PTB is associated with a less severe and transient neurobehavioral deficits

In the PCA30d group there was a milder sensory deficit noted on postnatal day 1 as compared to the PCA29d and PCA 28d (Fig 3). By 4 weeks of age in the PCA 30d less spontaneous alternation was seen in the TMT (Fig 4). Ultimately at 8 weeks of age in the PCA30d group, only mild behavior changes were noted in the OFT, with less digging and more self-grooming and rearing events (Table 1). The mild spatial memory insult at 4 weeks did not persist, but at 8 weeks of age some behavior indicative of anxiety was noted in the PCA30d group.

## Discussion

This small animal model instantiates neonatal and long-term neurobehavioral effects of iatrogenic prematurity in the absence of any contributing infective or hypoxic-ischemic insult. In these rabbits' prematurity resulted in a linear but inverse correlation between morbidity / mortality and PCA. In the earlier preterm groups, PCA28d and PCA29d, the insult resulted in a severe and persistent deficit while in the later PTB group, PCA30d, the neurobehavioral deficit was mostly transient and by 8w only mild anxiety driven behavior was noted compared to normal term controls.

A recent review specified criteria that could be used to define the utility of an animal model of preterm birth. The 'most useful' model should mimic what occurs in the clinical context of PTB, reflect the reproductive biology of humans, and have fetal maturation characteristics that

**Table 1. Open field behavior at 4 and 8 weeks of corrected postnatal age.**

| | Term birth PCA31d | Preterm PCA30d | Preterm PCA29d | Preterm PCA28d |
|---|---|---|---|---|
| Latency Time | | | | |
| • 4 weeks | 1 (1–3) | 2 (1–14) | 5 (2–11) | 6 (2–21) |
| • 8 weeks | 1 (1–7) | 1 (1–12) | 5 (1–47)* | 6 (1–50)* |
| Total Displacements | | | | |
| • 4 weeks | 34 (26–34) | 29 (28–38) | 42 (23–49) | 34 (27–44) |
| • 8 weeks | 30 (24–47) | 28 (21–46) | 32 (23–51) | 19 (11–28)** |
| Central Displacements | | | | |
| • 4 weeks | 8 (5–9) | 7 (5–10) | 5 (2–9) | 3 (2–6)** |
| • 8 weeks | 6 (4–8) | 7 (4–10) | 3 (2–7)** | 2 (1–4)*** |
| Exploration | | | | |
| • 4 weeks | 7 (6–9) | 8 (5–12) | 6 (3–7)* | 5 (2–8)* |
| • 8 weeks | 7 (5–9) | 7 (6–11) | 5 (3–7)** | 4 (2–7)*** |
| Standing Still | | | | |
| • 4 weeks | 3 (1–5) | 4 (2–7) | 7 (3–9))** | 6 (2–9)* |
| • 8 weeks | 2 (1–4) | 2 (1–6) | 6 (1–9)** | 6 (1–8)* |
| Resting | | | | |
| • 4 weeks | 0 (0–1) | 0 | 0 | 0 |
| • 8 weeks | 0 (0–1) | 0 (0–1) | 0 (0–1) | 0 (0–2) |
| Rearing | | | | |
| • 4 weeks | 2 (0–4) | 2 (0–10) | 6 (0–11)* | 2 (0–6) |
| • 8 weeks | 4 (3–8) | 6 (1–10)* | 8 (6–11)*** | 7 (0–11)* |
| Self-grooming | | | | |
| • 4 weeks | 1 (0–4) | 1 (0–3) | 2 (0–7) | 3 (1–5) |
| • 8 weeks | 1 (0–4) | 1 (0–1)* | 3 (1–4)* | 2 (1–4)* |
| Digging | | | | |
| • 4 weeks | 0 (0–2) | 0 (0–1) | 0 (0–1) | 0 (0–1) |
| • 8 weeks | 2 (0–3) | 0 (0–2)* | 0 (0–1)** | 0 (0–1)** |
| Biting | | | | |
| • 4 weeks | 0 (0–1) | 0 (0–1) | 1 (0–2) | 1 (0–2) |
| • 8 weeks | 0 (0–1) | 0 (0–1) | 1 (0–2) | 1 (0–2)** |
| Urinating | | | | |
| • 4 weeks | 0 | 0 (0–1) | 0 | 0 (0–1) |
| • 8 weeks | 0 (0–1) | 0 (0–1) | 0 (0–1) | 0 (0–1) |
| Defecating | | | | |
| • 4 weeks | 0 | 0 | 0 | 0 |
| • 8 weeks | 0 | 0 | 2 (0–3)** | 1 (1–3) |

PCA31d n = 11, PCA30d n = 11, PCA29d n = 7 and PCA28d n = 10. Data displayed as median and IQR with significance compared to the term birth group indicated as

* $0.05 \geq p > 0.01$

** $0.01 \geq p > 0.001$

***$p < 0.001$.

are similar to those of humans [28]. The rabbit is increasingly becoming a valuable experimental model in their own right and are in some cases the translational model of choice [17]. Rabbits are phylogenetically closer to primates than rodents and further offer a more diverse genetic background than in- and outbred rodent strains, which makes the model a better overall approximate to humans. Furthermore, due to the similarities that exist in placental and

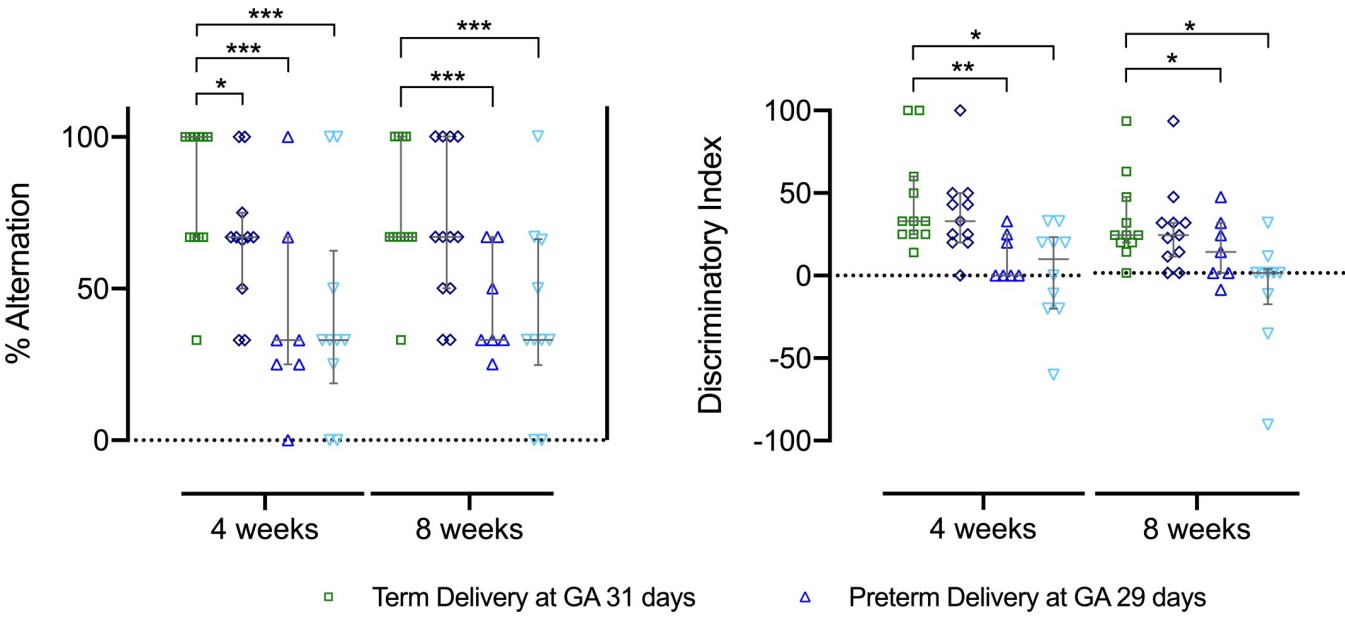

**Fig 4. Rabbit neurobehavioral assessment on corrected postnatal age of 4 and 8 weeks.** Spontaneous alternation percentage in the T-Maze test (A) and righting reflex events (B). PCA31d n = 11, PCA30d n = 11, PCA29d n = 7 and PCA28d n = 10. Data displayed as median and IQR with significance compared to the term birth group indicated as * $0.05 \geq p > 0.01$; ** $0.01 \geq p > 0.001$; ***$p < 0.001$.

perinatal cerebral development makes the rabbit an ideal model to examine prenatal insults in a controlled manner [17, 18]. Yet, to define the precise gestational age equivalents between humans and any animal is practically impossible since organs can have species specific independent developmental trajectories. While term born rabbits, PCA31d, can independently suckle, maintain their homeostasis and have strong enough locomotor capabilities to explore their nest, their brain [29] and lung development [30] are still more in keeping with a premature human infant. Furthermore, we have previously shown that rabbits born at 28 days shares more characteristics of severe or early PTB [19]. Ultimately the choice of one model over another depends on the purpose of the investigation.

The rabbit has also become a neuroscience model of choice being frequently used to evoke neurocognitive consequences of perinatal insults [31, 32]. Firstly this is motivated by the fact that rabbit's share similar brain structures to the human that are associated with learning and memory processes [33]. Secondly, their instinctive exploratory behavior with innate response to visual and olfactory stimuli makes the specie a perfect candidate for the behavioral neurosciences. Neurobehavioral assessments in rabbits have become widely accepted and the methodologies thereof have been greatly refined [34]. While animal models in behavioral neurosciences allows the study of brain-behavior relations under controlled conditions in living organisms, species-dependent behavior must be taken into account [35].

In this model there is apparent face validity, with clear parallels between the clinical context of prematurity related neurocognitive sequalae and neurobehavioral deficits in this PTB model. As in the clinical scenario the degree of prematurity directly correlates with the absolute risks of disability and death [36, 37]. The early preterm groups, PCA29 and PCA28d mimicked early or even extreme prematurity with high mortality and significant long term moderate to severe neurocognitive dysfunction [38]. While the mortality and morbidity seen in the late preterm group, PCA30d, was noticeably less. Yet, the distinct patterns in the PCA

30d resemble the clinical phenotype of late PTB. These rabbits had mild cognitive and behavioral deficits, with a small proportion of these rabbits resembling the outcomes of the early prematurity. Late preterm infants have a similar minor mortality risk with a mild risk of behavioral, cognitive and social-emotional competence difficulties, but a small proportion of these infants had the same outcome profile of early premature infant with a high risk of behavioral and socio-emotional problems [39].

Overall, this model can now be further exploited to investigate underlying neuropathological mechanisms. Especially the late PTB model, PCA30d, could be used to clarify some research questions typically encountered in this late preterm period. For example, there is still a heated debate around the question on how to address growth restriction [40] during this specific gestation and subsequently the infant respiratory [41] and neonatal morbidity [10, 42].

The strengths of this model rest in the strict application of neurobehavioral assessment methodology and the blinded review thereof. Furthermore, through the utilization of intra- and inter-litter randomization combined with cross fostering we aimed to control the maternal and litter associated biases. There is no general consent on how to weight a model's validity but by restricting the methodology to stringent criteria, we aimed to boost the reliability and replicability of the model. Moreover, the constructive validity seems plausible as the underlying interaction between PTB and prematurity related neurocognitive deficit has been well documented in the rabbit [19] and these results are in keeping with other models of PTB related prematurity deficits [28].

Lastly in this model a surgical preterm delivery was chosen since the goal was to determine the degree of prematurity that would result in a cognitive and behavioral morbidity phenotype that would parallel the clinical preterm infant scenario. This was done not because this is the most common delivery method in preterm deliveries, but in order to avoid the confounders of labor and intrapartum stress. Thereby avoiding an inflammation mediated PTB. Although inflammation is seen as a common trigger of early and late preterm birth [43] this brings about another contributory insult. On its own inflammation can negatively impact the neonate [44] and a clear association exists between perinatal inflammation, neuroinflammation and poor neurocognitive outcomes [45].

The complexity of prematurity in humans can never be entirely replicated in an animal model, consequently several limitations were encountered in this translational work. As we have noted previously the biggest concern is that specific inter-specie GA equivalents are practically impossible due to the multitude of specie specific dynamic biological development processes [19]. Herein we generalized that at the one end of the spectrum PCA28d bear a resembles to early severe PTB while PCA30d that of late PTB simply based upon the general health and neurobehavioral outcomes alone.

Specifically, the mortality seen in this model was ascribed to prematurity with the inability of vital organ functions to sustain life, since no clear causations could be established. Yet, pre-emptive empiric interventions to address this mortality could add another confounder to this study. Thereby, derailing the purpose of this work to investigate the role of GA alone in this rabbit model. Future work will focus on the neuropathological footprint in this model and the underlying mechanisms that effect the neurobehavioral outcomes. Lastly, other reported confounders that could have influenced these outcomes such as maternal stress, sex, systemic inflammation and gut health should be considered in forthcoming research [28].

In conclusion, in this rabbit model, delivery at PCA30d resembles the clinical context of late PTB. These rabbits suffer from a clear transient insult with the majority of them being similar to their term born mates while some had persistent mild behavioral deficits noted. This rabbit model enables the possibility to evaluate various important perinatal insults that commonly occur in the late preterm period.

## Supporting information

**S1 Table. Maternal observation at delivery and litter outcomes.** PCA31d n = 4, PCA30d n = 4, PCA29d n = 5, PCA28d n = 6 and PCA27d n = 2. Data displayed as mean and SD with significance compared to the term birth group indicated as * $0.05 \geq p > 0.01$; ** $0.01 \geq p > 0.001$; ***p $< 0.001$.
(DOCX)

**S2 Table. Postnatal day 1 evaluation, PND1.** PCA31d n = 11, PCA30d n = 11, PCA29d n = 8 and PCA28d n = 11. Data displayed as mean and SD with significance compared to the term birth group indicated as * $0.05 \geq p > 0.01$; ** $0.01 \geq p > 0.001$; ***p $< 0.001$.
(DOCX)

**S3 Table. NORT—Novel object recognition test at 4 and 8 weeks of corrected postnatal age.** PCA31d n = 11, PCA30d n = 11, PCA29d n = 7 and PCA28d n = 10. Data displayed as mean and SD with significance compared to the term birth group indicated as * $0.05 \geq p > 0.01$; ** $0.01 \geq p > 0.001$; ***p $< 0.001$.
(DOCX)

**S1 File. Neurobehavioral examinations.**
(DOCX)

## Author Contributions

**Conceptualization:** Johannes van der Merwe, Jaan Toelen, Jan Deprest.

**Data curation:** Johannes van der Merwe, Lennart van der Veeken, Analisa Inversetti, Angela Galgano, Jan Deprest.

**Formal analysis:** Johannes van der Merwe, Lennart van der Veeken, Analisa Inversetti, Jaan Toelen, Jan Deprest.

**Funding acquisition:** Jan Deprest.

**Investigation:** Johannes van der Merwe.

**Methodology:** Johannes van der Merwe, Jaan Toelen, Jan Deprest.

**Project administration:** Johannes van der Merwe.

**Resources:** Johannes van der Merwe, Jaan Toelen, Jan Deprest.

**Supervision:** Jaan Toelen, Jan Deprest.

**Visualization:** Johannes van der Merwe.

**Writing – original draft:** Johannes van der Merwe, Jaan Toelen, Jan Deprest.

**Writing – review & editing:** Johannes van der Merwe, Lennart van der Veeken, Analisa Inversetti, Angela Galgano, Jaan Toelen, Jan Deprest.

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
