## [Decision Letter · Decision Letter 0]

4 Dec 2020

PONE-D-20-30953

Earlier preterm birth is associated with a worse neurocognitive outcome in a rabbit model.

PLOS ONE

Dear Dr. Johannes Van der Merwe,

Thank you for submitting your manuscript to PLOS ONE. After careful consideration, we feel that it has merit but does not fully meet PLOS ONE’s publication criteria as it currently stands. Therefore, we invite you to submit a revised version of the manuscript that addresses the points raised during the review process.

We look forward to receiving your revised manuscript.

Kind regards,

Kazumichi Fujioka

Academic Editor

PLOS ONE

Journal Requirements:

3.Thank you for stating the following in the Funding Section of your manuscript:

[JvdM and LvdV are funded with support of the Erasmus + Programme of

the European Union (Framework Agreement number: 2013-0040). This publication

reflects the views only of the author, and the Commission cannot be held responsible

for any use which may be made of the information contained therein.]

 [The funders had no role in study design, data collection and analysis, decision to publish, or preparation of the manuscript.]

5. As part of your revisions, please provide mortality rate for the animals during your study (if applicable) and also provide monitoring parameters and details about the use of humane endpoints. If there were any unexpected adverse events, please report on those, as well. Thank you!

6.We noticed you have some minor occurrence of overlapping text with the following previous publication, which needs to be addressed:

https://www.nature.com/articles/s41598-019-39922-8?code=7ae1192a-cb9b-4c6a-b8e3-c693855b4ffa&error=cookies_not_supported

The text that needs to be addressed involves lines 415-426.

In your revision ensure you cite all your sources (including your own works), and quote or rephrase any duplicated text outside the methods section. Further consideration is dependent on these concerns being addressed.

Additional Editor Comments (if provided):

The paper is interesting, however; the authors should reply to the critiques especially arisen by reviewer 2. The issue of positive and negative control is important. In addition, in clinical setting, the reason of pregnancy termination such as CAM or threatened preterm labor could be affecting the perinatal and long term outcome. The authors might want to discuss with this. And as a interest, the reason of mortality in the preterm group is for what? respiratory distress syndrome??

Reviewers' comments:

Reviewer's Responses to Questions

**Comments to the Author**

1. Is the manuscript technically sound, and do the data support the conclusions?

Reviewer #1: Yes

Reviewer #2: Partly

2. Has the statistical analysis been performed appropriately and rigorously? 

Reviewer #1: Yes

Reviewer #2: Yes

3. Have the authors made all data underlying the findings in their manuscript fully available?

Reviewer #1: Yes

Reviewer #2: Yes

4. Is the manuscript presented in an intelligible fashion and written in standard English?

Reviewer #1: Yes

Reviewer #2: Yes

5. Review Comments to the Author

Reviewer #1: Van der Merwe and colleagues provide here a rabbit model that reflects the clinical phenotypes of early and late preterm birth. By using caesarean delivery at a postconceptional age (PCA) of either 28, 29, 30 or 31 days, the author aimed to provide a model to investigate perinatal insults during the early and late preterm period. Newborn rabbits were mixed and randomly allocated to be raised by cross fostering and underwent neurobehavioral testing. The earlier preterm groups, PCA28d and PCA29d, mimicked early or extreme prematurity with high mortality and neurocognitive dysfunctions. In the later preterm group, the PCA30d, showed transient neurobehavioral deficit and mild anxiety at 8 weeks. They concluded that this model could be used to investigate underlying pathological mechanism of perinatal insults during the early and late preterm period.

This is a nice description of a new model to investigate clinical phenotypes of early and late preterm birth. The experimental procedures were well designed and the results are clear. The manuscript is properly written and the conclusions are well supported by the results.

Minor comments;

1. Line 131, the authors described, “for both short-term and long-term groups 10 kittens from each delivery gestation underwent neurobehavioral testing”. Line 280, “In the PCA31d, PCA30d and PCA28d groups, 11 rabbits were evaluated at each time point while in the PCA29d group 8 rabbits were evaluated due to smaller litter size and survival rates.” These descriptions are a bit confusing. How many kittens were actually evaluated? In addition, this reviewer prefers to describe actual numbers of evaluated animals in the Figure Titles, as like Table 3.

2. Suppl 2, Additonal Data, Table 12 might be Table 2.

Reviewer #2: Thank you for the opportunity to review this manuscript by Merwe et al. Here the authors developed the early and late preterm birth model in the rabbit. The study and established model are important to understand the mechanism of why preterm birth affects the behavior or emotion of a baby, especially late-preterm infants. However, I have some concerns that should be addressed for revision.

1. Have the authors considered the litter bias and sex in this study? Can you write it in the text?

2. If the author wants to insist that you have developed the model, I think that you should show and compare the positive control or the negative control, or both, with each experiment. Have the authors any data?

3. Has the author conducted a pathological analysis of the brain? Can you show us the HE stained samples? In Fig. 2C, there is a significant reduction in brain weight in PD at GA28 days group. What do you think of that?

4. In line 347-350. The author insists that “Rabbits are phylogenetically closer to primates than rodents and further offer a more diverse genetic background than in-and outbred rodent strains, which makes the model a better overall approximate to humans”. Can the author provide any evidence or quote references? This description may be exaggerated.

5. I can't understand why the authors chose a rabbit for this model rather than a mouse. If the author wants to apply this model to elucidate the mechanism, I think it would be much easier to develop research in mice. Because the mouse is easier to experiment with biochemistry and molecular biological analysis than the rabbit. Please provide some more discussion about the benefits of studying rabbits.

6. In line 124, please correct Oryctolagus cuniculus in italics.

6. PLOS authors have the option to publish the peer review history of their article (what does this mean?). If published, this will include your full peer review and any attached files.

Reviewer #1: No

Reviewer #2: No

---

## [Author Response · Author response to Decision Letter 0]

4 Jan 2021

We are pleased with the detailed review of our manuscript entitled “Earlier preterm birth is associated with a worse neurocognitive outcome in a rabbit model”. The suggested amendments will most certainly improve the quality of the manuscript. 

For this response, we have replied in numbered style in the order of the comments made by the reviewer. 

1. Editor Comments

1.1. ‘Please ensure that your manuscript meets PLOS ONE's style requirements, including those for file naming. The PLOS ONE style templates can be found at

https://journals.plos.org/plosone/s/file?id=wjVg/PLOSOne_formatting_sample_main_body.pdf and https://journals.plos.org/plosone/s/file?id=ba62/PLOSOne_formatting_sample_title_authors_affiliations.pdf’

 The manuscript’s style formatting has been edited as per above instructions. 

1.2. ‘We note that you have included the phrase “data not shown” in your manuscript. Unfortunately, this does not meet our data sharing requirements. PLOS does not permit references to inaccessible data. We require that authors provide all relevant data within the paper, Supporting Information files, or in an acceptable, public repository. Please add a citation to support this phrase or upload the data that corresponds with these findings to a stable repository (such as Figshare or Dryad) and provide and URLs, DOIs, or accession numbers that may be used to access these data. Or, if the data are not a core part of the research being presented in your study, we ask that you remove the phrase that refers to these data.’

We have added a reference (no 21) for this specific data, and we have also included the data from a trial that we have done of PCA27d deliveries in the S2 Table 1. 

1.3. ‘Thank you for stating the following in the Funding Section of your manuscript:

[JvdM and LvdV are funded with support of the Erasmus + Programme of

the European Union (Framework Agreement number: 2013-0040). This publication

reflects the views only of the author, and the Commission cannot be held responsible

for any use which may be made of the information contained therein.]

 [The funders had no role in study design, data collection and analysis, decision to publish, or preparation of the manuscript.]

Please include your amended statements within your cover letter; we will change the online submission form on your behalf.’

The funding statement has been removed from the manuscript and added to the cover letter as instructed. It should read as follows:

JvdM and LvdV are funded with support of the Erasmus + Programme of the European Union (Framework Agreement number: 2013-0040). This publication reflects the views only of the author, and the Commission cannot be held responsible for any use which may be made of the information contained therein. The funders had no role in study design, data collection and analysis, decision to publish, or preparation of the manuscript. 

1.4. ‘Please include captions for your Supporting Information files at the end of your manuscript, and update any in-text citations to match accordingly. Please see our Supporting Information guidelines for more information: http://journals.plos.org/plosone/s/supporting-information.’

The manuscript has been amended and the supporting information files captions have been added to the manuscript. 

1.5. ‘As part of your revisions, please provide mortality rate for the animals during your study (if applicable) and also provide monitoring parameters and details about the use of humane endpoints. If there were any unexpected adverse events, please report on those, as well.’

The stillbirth rate is given in S2 Table 1 and the overall mortality rate is given in Fig 2A. A description of animal care was added to the methods (line 112 and 122-123). 

1.6. ‘We noticed you have some minor occurrence of overlapping text with the following previous publication, which needs to be addressed:

https://www.nature.com/articles/s41598-019-39922-8?code=7ae1192a-cb9b-4c6a-b8e3-c693855b4ffa&error=cookies_not_supported

The text that needs to be addressed involves lines 415-426.’

Yes, there is some overlapping with our previous publication in Nature. After discussion we have decided to rewrite the limitations section in order to highlight the shortcomings of this work as noted by the reviewers (line 434-447). 

1.7. ‘In addition, in clinical setting, the reason of pregnancy termination such as CAM or threatened preterm labor could be affecting the perinatal and long term outcome. The authors might want to discuss with this. And as a interest, the reason of mortality in the preterm group is for what? respiratory distress syndrome??'’

As pointed out in the intrioduction the aim of this model is to characterise the effects of prematurity alone, i.e. that is not influenced by other confounders and additional insults (line 103-107). 

2. Reviewer #1: 

2.1. “Line 131, the authors described, “for both short-term and long-term groups 10 kittens from each delivery gestation underwent neurobehavioral testing”. Line 280, “In the PCA31d, PCA30d and PCA28d groups, 11 rabbits were evaluated at each time point while in the PCA29d group 8 rabbits were evaluated due to smaller litter size and survival rates.” These descriptions are a bit confusing. How many kittens were actually evaluated? In addition, this reviewer prefers to describe actual numbers of evaluated animals in the Figure Titles, as like Table 3”

Correct this is misleading. The number never remained constant for each subgroup as we had many readouts in both the short and long term groups. As correctly pointed out we have added the absolute number to each figure and table and this sentece was removed from the method section (line 120). 

2.2. “Suppl 2, Additional Data, Table 12 might be Table 2.”

Thank you for spotting this. The title was amended. 

3. Reviewer #2: 

3.1. “Have the authors considered the litter bias and sex in this study? Can you write it in the text?”

Yes, as pointed out in the methods we tried to limit the litter bias by randomizing and mixing the litters and in addition we used cross-fostering (line 144-150). We did not consider the influence of sex in this study and added this as a limitation in the discussion (line 447-448), this is an interesting point, and we would consider including the sex determination in future experiments 

3.2. “If the author wants to insist that you have developed the model, I think that you should show and compare the positive control or the negative control, or both, with each experiment. Have the authors any data?”

Shortly no, not because we didn’t consider this but selecting a true control group was just not possible. A prefect negative control could not be “created”, even if we allowed the rabbits to deliver spontaneously at term then we would not be able to compare the influence of the surgical delivery. Therefore, all rabbits were delivered surgically but at different gestations with the term born group seen as a ‘negative control’. This was highlighted in the discussion (line 423-427).

3.3. “Has the author conducted a pathological analysis of the brain? Can you show us the HE stained samples? In Fig. 2C, there is a significant reduction in brain weight in PD at GA28 days group. What do you think of that?”

Correct, the next step is to objectify the insult further with neuropathological and MRI analysis as pointed out in line. This forms part of a subsequent research project and the results will take about another year. 

3.4. “In line 347-350. The author insists that “Rabbits are phylogenetically closer to primates than rodents and further offer a more diverse genetic background than in-and outbred rodent strains, which makes the model a better overall approximate to humans”. Can the author provide any evidence or quote references? This description may be exaggerated.”

Correct, no single animal can substitute the human condition, but we motivated our rabbit model use in the introduction with a reference to rabbit brain development (line 94-97) and added to the discussion in the limitations (line 436-438).

3.5. “I can't understand why the authors chose a rabbit for this model rather than a mouse. If the author wants to apply this model to elucidate the mechanism, I think it would be much easier to develop research in mice. Because the mouse is easier to experiment with biochemistry and molecular biological analysis than the rabbit. Please provide some more discussion about the benefits of studying rabbits.”

Correct, as pointed out above no single model is absolutely perfect no matter which model a research line chooses as pointed out in the introduction and discussion. 

3.6. “In line 124, please correct Oryctolagus cuniculus in italics.”

Thank you for noting this, it was corrected.

---

## [Decision Letter · Decision Letter 1]

12 Jan 2021

Earlier preterm birth is associated with a worse neurocognitive outcome in a rabbit model.

PONE-D-20-30953R1

Dear Dr.Johannes Van der Merwe,

We’re pleased to inform you that your manuscript has been judged scientifically suitable for publication and will be formally accepted for publication once it meets all outstanding technical requirements.

Kind regards,

Kazumichi Fujioka

Academic Editor

PLOS ONE

Additional Editor Comments (optional):

Reviewers' comments:

Reviewer's Responses to Questions

**Comments to the Author**

1. If the authors have adequately addressed your comments raised in a previous round of review and you feel that this manuscript is now acceptable for publication, you may indicate that here to bypass the “Comments to the Author” section, enter your conflict of interest statement in the “Confidential to Editor” section, and submit your "Accept" recommendation.

Reviewer #1: All comments have been addressed

Reviewer #2: All comments have been addressed

2. Is the manuscript technically sound, and do the data support the conclusions?

Reviewer #1: Yes

Reviewer #2: Yes

3. Has the statistical analysis been performed appropriately and rigorously? 

Reviewer #1: Yes

Reviewer #2: I Don't Know

4. Have the authors made all data underlying the findings in their manuscript fully available?

Reviewer #1: Yes

Reviewer #2: Yes

5. Is the manuscript presented in an intelligible fashion and written in standard English?

Reviewer #1: Yes

Reviewer #2: Yes

6. Review Comments to the Author

Reviewer #1: The authors have responded to my concerns properly. Their model will be useful to investigate perinatal insults. I have no further concerns about this article.

Reviewer #2: Thank you for the opportunity to review this manuscript by Merwe et al. As the authors understand, it is important to set positive or negative control at all times. I recommend you should use an anti-depressant such as desipramine when you conduct a behavior test as a positive control to prove that the experiment has been done correctly.

7. PLOS authors have the option to publish the peer review history of their article (what does this mean?). If published, this will include your full peer review and any attached files.

Reviewer #1: No

Reviewer #2: No

---

## [Editor Report · Acceptance letter]

18 Jan 2021

PONE-D-20-30953R1 

Earlier preterm birth is associated with a worse neurocognitive outcome in a rabbit model. 

Dear Dr. Van der Merwe:

I'm pleased to inform you that your manuscript has been deemed suitable for publication in PLOS ONE. Congratulations! Your manuscript is now with our production department. 

Kind regards, 

on behalf of

Dr. Kazumichi Fujioka 

Academic Editor

PLOS ONE